# “Vaccine Passports” May Backfire: Findings from a Cross-Sectional Study in the UK and Israel on Willingness to Get Vaccinated against COVID-19

**DOI:** 10.3390/vaccines9080902

**Published:** 2021-08-14

**Authors:** Talya Porat, Ryan Burnell, Rafael A. Calvo, Elizabeth Ford, Priya Paudyal, Weston L. Baxter, Avi Parush

**Affiliations:** 1Dyson School of Design Engineering, Imperial College London, London SW7 2DB, UK; r.burnell@imperial.ac.uk (R.B.); r.calvo@imperial.ac.uk (R.A.C.); weston.baxter@imperial.ac.uk (W.L.B.); 2Department of Primary Care and Public Health, Brighton and Sussex Medical School, Brighton BN1 9PH, UK; E.M.Ford@bsms.ac.uk (E.F.); P.Paudyal@bsms.ac.uk (P.P.); 3The Faculty of Industrial Engineering & Management, Technion Israel Institute of Technology, Haifa 3200003, Israel; aparush@technion.ac.il

**Keywords:** COVID-19, public health, self-determination theory, vaccine passports, vaccination

## Abstract

Domestic “vaccine passports” are being implemented across the world as a way of increasing vaccinated people’s freedom of movement and to encourage vaccination. However, these vaccine passports may affect people’s vaccination decisions in unintended and undesirable ways. This cross-sectional study investigated whether people’s willingness and motivation to get vaccinated relate to their psychological needs (autonomy, competence and relatedness), and how vaccine passports might affect these needs. Across two countries and 1358 participants, we found that need frustration—particularly autonomy frustration—was associated with lower willingness to get vaccinated and with a shift from self-determined to external motivation. In Israel (a country with vaccine passports), people reported greater autonomy frustration than in the UK (a country without vaccine passports). Our findings suggest that control measures, such as domestic vaccine passports, may have detrimental effects on people’s autonomy, motivation, and willingness to get vaccinated. Policies should strive to achieve a highly vaccinated population by supporting individuals’ autonomous motivation to get vaccinated and using messages of autonomy and relatedness, rather than applying pressure and external controls.

## 1. Introduction

Since the start of the COVID-19 pandemic, the world’s hopes have been pinned on vaccines that have the potential to allow people to return to “life as normal” [1]. The rollout of these vaccines is now well underway, with 2.3 billion people (29.6% of the world’s population) having received at least one dose at the time of writing. Some countries have already vaccinated the majority of their populations—for instance, 62.3% of Israel’s population are fully vaccinated, as is 57.3% of the UK’s population [2].

These high proportions demonstrate that people in most countries, on the whole, have accepted the need for vaccines and are eager to get vaccinated [3,4]. However, there remain some individuals who are reluctant to take the vaccine. In Israel, for example, 15% of the eligible adult population have not taken up the opportunity to get vaccinated [5]. Likewise, 11% of eligible adults in the UK have not yet chosen to get vaccinated [6]. Although these reluctant groups are relatively small, they are not insignificant—some estimates suggest that any vaccine refusal rate greater than 10% could significantly hinder herd immunity [1]. Therefore, if we want to ensure enough people choose to get vaccinated to control the spread of the virus, it is vital that we understand the factors that affect people’s willingness to get vaccinated.

One important policy that might affect willingness to get vaccinated is vaccine passports. In order to allow vaccinated individuals to move freely and return to daily activities, several countries have introduced or considered measures that would restrict access to public spaces for people who are unvaccinated [7]. The first country to implement such a policy was Israel. “Green passes” were provided from January 2021 onwards to fully vaccinated residents or people who had recovered from COVID-19, permitting entry to otherwise restricted sites such as gyms, restaurants, hotels, theatres, and music venues. In the US, New York required vaccine certification in order to access certain social activities, and many other states have also expressed interest in the idea [8]. In Europe, Denmark launched its “coronapas” system in March to be used domestically [9].

The goal of vaccine passports is to pave the way for economic recovery and restore people’s freedoms [10]. However, these passports also raise concerns about potential violations of people’s autonomy and freedom of choice [7,11,12]. For example, in the UK, more than 375,000 people signed a petition against the rollout of COVID-19 vaccine passports because they “could be used to restrict the rights of people who have refused a COVID-19 vaccine” (https://petition.parliament.uk/petitions/569957, accessed on 8 August 2021). Even putting aside these ethical issues, it remains unclear how these passports might affect people’s vaccination decisions and well-being. On the one hand, vaccine passports could incentivize people to get vaccinated so they are able to move freely in society [11]. However, on the other hand, there are reasons to think that measures such as vaccine passports might actually increase some people’s resistance to vaccination or alter the motivation behind their decision to get vaccinated in ways that might have detrimental long-term consequences.

Decades of research shows that societies and individuals can only flourish in environments that foster basic psychological needs [13]. According to self-determination theory (SDT), there are three of these needs: a need for autonomy (a sense of meaning, volition, choice over one’s life), a need for competence (the feeling of being capable of achieving one’s goals and overcoming challenges), and a need for relatedness (feeling cared for by others, trusted and understood). Satisfaction of these three psychological needs is critical for self-regulating and sustaining behaviours that improve health and well-being, such as exercising, smoking cessation and adherence to prescribed medications [14,15]. Recent evidence also suggests that the satisfaction of these needs is important for adherence to preventative COVID-19 measures [16,17]. In contrast, the frustration of these needs may elicit ill-being, a lack of motivation to act, or in some cases might even provoke defensiveness (doing the opposite of what is requested) [18,19]. People with frustrated needs are also more drawn to conspiracy theories, which could feed into vaccine hesitancy [20,21].

Together, these data provide reasons to expect that people’s willingness and motivation to get vaccinated will depend on their psychological needs—more specifically, the extent to which they feel a sense of autonomy over the decision to get vaccinated, the extent to which they feel competent in their ability to get vaccinated, and the extent to which they feel a sense of relatedness to local and health authorities. If this hypothesis is correct, then if vaccine passports frustrate people’s psychological needs—for example by making people feel a lack of autonomy over their decision—then these passports might paradoxically reduce people’s willingness to get vaccinated.

Furthermore, behaviours are more likely to be sustained over time if people’s motivation for engaging in those behaviours is self-determined and autonomous (performed for internal reasons) than if their motivation is controlled (performed due to external pressures) [18,22]. In addition, there is evidence that the frustration of psychological needs can shift people’s motivation from autonomous to controlled [23,24].

A common form of autonomous motivation is *identified regulation*—when one identifies and understands the value and importance of a behaviour. This is facilitated when local authorities provide meaningful rationales for a behaviour, and do not apply pressure and external controls [14]. In contrast, common forms of controlled motivation are *external regulation*, in which one only acts to avoid punishment, receive a reward or be in accordance with social pressure and *introjected regulation*, in which one acts to receive approval or avoid feelings of guilt [14]. According to SDT, in contrast to autonomous motivation, these forms of controlled regulations are not sustainable and may improve adherence only for a short period of time [25]. In the context of vaccination, measures such as vaccine passports may increase vaccination uptake in the short term, but might also shift people’s motivation to external or introjected, making them less likely to sign-up for a second dose of the vaccine, less willing to take up the opportunity to receive a “booster” shot, or less willing to take a yearly vaccine against new variants.

Given these potential detrimental effects of vaccine passports, the aim of this study was to investigate whether people’s willingness and motivation to get vaccinated depends on their psychological needs, and how vaccine passports might affect these needs. Recent studies have called to evaluate the unintended secondary negative effects of vaccine passports, in addition to their effectiveness and impact [7]. This is the first study to our knowledge to investigate the unintended consequences of domestic vaccine passports using self-determination theory. The results have implications for policy decisions regarding vaccine passports and will help in understanding the importance of autonomy, competence and relatedness in people’s vaccination decisions. We collected data from two countries, one that has implemented vaccine passports and one that has not—Israel and the UK, respectively. We asked participants to report the extent to which their psychological needs were satisfied and frustrated in relation to getting vaccinated. Then, we asked them to report whether they were vaccinated, how willing they would be to get vaccinated and what their attitudes were towards vaccine passports.

## 2. Materials and Methods

### 2.1. Design and Setting

The pre-registration for the study is available at: osf.io/vtz7h. The study was an online survey disseminated online via Prolific in the UK [26] and via PanelView in Israel [27]. Data collection began on the 10 May 2021 and ended on the 14 May 2021. Israel and the UK were selected for this study because at the time of the study (10 May, 2021), they were the two leading countries in terms of vaccination rate (Israel was leading with 62.7% of its population having received at least one dose, followed by the UK with 52.4% [2]). The fact that many people in these countries would have had the opportunity to get vaccinated allowed us to examine predictors of actual vaccination decisions in addition to intentions.

### 2.2. Participants

In line with our pre-registration, 1411 participants completed the study (701 from the UK and 710 from Israel). Among them, we excluded 20 participants from the UK and 33 from Israel who failed the attention check, leaving us with our final sample of 1358 participants (681 from the UK and 677 from Israel). Both samples were representative of the country’s demographics. All participants were aged 18 or older. Participants received GBP 1.4–GBP 1.6 for their participation.

### 2.3. Main Outcome Measures

#### 2.3.1. Psychological Need Satisfaction and Frustration

We adapted 12 items from the Basic Psychological Need Satisfaction and Frustration Scale (BPNSNF) [28] to the context of getting vaccinated, with two items assessing each of the six constructs—autonomy satisfaction; autonomy frustration; relatedness satisfaction; relatedness frustration; competence satisfaction; and competence frustration (See Table 1). Each item was rated from 1 (strongly disagree) to 5 (strongly agree). The Hebrew translation was based on Benita et al. [29].

#### 2.3.2. Vaccination Behaviour 

We asked participants whether they were vaccinated, and if so, how many doses they received.

#### 2.3.3. Willingness to Get vaccinated

Our main dependent measure was participants’ willingness to get vaccinated, which we measured in two ways. First, we asked people how willing they are (or were, if they have already been vaccinated) to get vaccinated from 1 (not at all willing) to 5 (extremely willing). Then, we asked people who were not yet vaccinated whether they would choose to get vaccinated (yes/no). 

#### 2.3.4. Attitudes towards “COVID Passports”

Attitudes towards COVID passports were measured by asking participants the extent to which they support three scenarios: A “COVID passport” enabling only people who got *fully vaccinated* to perform some activities (e.g., stay in hotels, participate in large events, etc.); A “COVID passport” enabling people who got *fully vaccinated or recently tested* to perform some activities (e.g., stay in hotels, participate in large events, etc.); and mandatory vaccination for all residents. 

#### 2.3.5. Motivations to Get Vaccinated/Not to Get Vaccinated

We measured different motivations to get vaccinated/not to get vaccinated using the Treatment Self-Regulation Questionnaire (TSRQ) [30,31], with two items for each kind of motivation, measured from 1 (Not at all true) to 7 (Very true), as can be seen in Table 2.

#### 2.3.6. Demographics 

We asked participants to report their age, gender, religion, education, employment status, whether they have children, marital status, ethnicity and county, as can be seen in Table 3.

### 2.4. Power Calculation

This sample size provides us with greater than 99% power to detect small effects (f^2^ = 0.05) in the regression investigating the relationship between the six psychological need variables and people’s willingness to get vaccinated against COVID-19.

### 2.5. Statistical Analysis

In line with our pre-registration, for our primary analysis, we conducted a linear regression with the six psychological needs (autonomy satisfaction, autonomy frustration, competence satisfaction, competence frustration, relatedness satisfaction, and relatedness frustration) as predictors and people’s willingness to get vaccinated as the dependent measure. We also conducted a logistic regression with the same predictors, with the dichotomous intention to get vaccinated measure as the dependent measure. We investigated the possible effects of vaccine passports on willingness to get vaccinated by calculating the mean difference in need frustration between Israeli and UK participants and the 95% confidence interval around that difference. 

### 2.6. Pilot Studies

Two pilot studies, one with 100 participants (50 from each country on April 29) and one with 60 participants (30 from each country on May 5) were performed through Prolific to receive feedback from members of the public about the survey. Based on the feedback, changes to the questionnaire (improving clarity of the questions, removing and adding questions) were made after each pilot.

## 3. Results

### 3.1. Demographics, Vaccination Status and Willingness to Get Vaccinated

Demographics and vaccination status of participants from the UK and Israel is displayed in Table 3. Among the 229 participants in the UK who were unvaccinated, 69% (or 23.05% of the full UK sample) said they intended to get vaccinated, while 31% (or 10.57% of the full UK sample) said they did not intend to get vaccinated. Less than 0.5% reported they could not get vaccinated due to medical reasons. Of the 97 participants in Israel who were unvaccinated, 19% (or 2.66% of the full Israel sample) said they intended to get vaccinated, while 81% (or 11.67% of the full Israel sample) said they did not. Only 1% reported they could not get vaccinated due to medical reasons. 

In terms of willingness to get vaccinated (rated from 1—not at all willing to 5—extremely willing), participants in the UK who had already received at least one dose of the vaccine reported being highly willing to get vaccinated (*Mdn* = 5, *M* = 4.71, *SD* = 0.63). Participants in Israel who had received at least one dose were also relatively willing, although less so than people in the UK (*Mdn* = 4, *M* = 4.06, *SD* = 1.14). Among the participants who had not yet been vaccinated, participants in the UK were relatively willing to do so (*Mdn* = 4, *M* = 3.72, *SD* = 1.49), but participants in Israel tended to be reluctant (*Mdn* = 2, *M* = 2.15, *SD* = 1.24).

### 3.2. Psychological Needs

Our primary research aim was to investigate the relationships between people’s psychological needs and their willingness to get vaccinated. To do so, we conducted a linear regression with the six need ratings predicting people’s willingness to get vaccinated. As Table 4 shows, autonomy frustration was the strongest predictor of people’s willingness to get vaccinated, such that the more people felt autonomy frustrated (forced to get vaccinated or “punished” if not), the less willing they were to get vaccinated. In addition, autonomy satisfaction and relatedness satisfaction also predicted people’s willingness to get vaccinated such that the more people felt volition and choice and that the authorities care about and understand their needs, the more willing they were to get vaccinated. 

Ultimately, however, people have to make a decision one way or the other about whether to get vaccinated, so it is useful to consider whether psychological needs are related to these dichotomous decisions. To do so, we conducted a logistic regression using a dichotomous “decision to get vaccinated” variable we created, with anyone who has had at least one dose of the vaccine or who said that they would choose to get vaccinated coded as a 1, and anyone who said they would not choose to get vaccinated coded as a 0. This regression (see Table 5) showed a very similar pattern—once again, autonomy frustration, autonomy satisfaction and relatedness satisfaction were by far the strongest predictors of intentions to get vaccinated. Together, these relationships are consistent with the hypothesis that psychological needs affect people’s willingness to get vaccinated.

Of course, vaccination intentions do not necessarily reflect people’s vaccination behaviour. Therefore, we next examined how psychological needs relate to whether people had received at least one dose of the vaccine. The results were similar to those of the vaccination intention analyses—we found that autonomy frustration (*β* = −0.19, *p* = 0.022), and relatedness satisfaction (*β* = 0.58, *p* = <0.001) predicted people’s vaccination status, as did competence satisfaction (*β* = 0.34, *p* = 0.004).

Then, we sought to investigate the possibility that need frustration around vaccination might affect people’s motivation to get vaccinated. To do so, we investigated the relationships between need frustration and four types of motivation: identified, introjected, external, and amotivation. 

Across all three needs, we found that need frustration was negatively correlated with identified motivation (i.e., understanding the value and importance of getting vaccinated). In addition, autonomy and relatedness frustration were negatively correlated with introjected motivation (acting to receive approval or avoid feelings of guilt). These findings suggest that people who feel that their autonomy, competence, or relatedness are frustrated are less likely to have self-determined motivation to get vaccinated. Frustration of each need was also positively correlated with amotivation, and frustration of autonomy and competence were positively correlated with external motivation (acting only to avoid punishment or conform to social pressure). Together, these findings suggest that people whose psychological needs are frustrated tend to be more externally motivated to get vaccinated and care less about getting vaccinated. These correlations are displayed in Table 6.

The findings we described thus far suggest that need frustration is related to both an unwillingness to get vaccinated and a shift from self-determined to external motivation. Therefore, to the extent that vaccine passports frustrate people’s psychological needs, these passports might have undesirable effects on people’s vaccination behaviour and motivation. However, do vaccine passports frustrate psychological needs? To address this question, we compared the experience of psychological needs in a country with vaccine passports (Israel) to a country without them (United Kingdom). We found that autonomy frustration was markedly higher in Israel than in the UK, Mdiff = 0.62, 95% CI [0.50, 0.74]). Both competence and relatedness frustration were also higher in Israel, although the difference between the countries was smaller than for autonomy, Mdiff(competence) = 0.24, 95% CI [0.15, 0.33]; Mdiff(relatedness) = 0.39, 95% CI [0.29, 0.50] (see Figure 1). This hypothesis is further supported by the relationships between support for vaccine passports and need frustration. For example, the more participants were against vaccine passports, the more autonomy frustration they reported: r(1356) = −0.37, 95% CI [−0.32, −0.41].

## 4. Discussion

Across two countries and 1358 participants, we investigated the relationship between psychological needs and people’s motivation and willingness to take the COVID-19 vaccine. In both the UK and Israel, we found that need frustration—particularly autonomy frustration—predicted unwillingness to get vaccinated and a shift from self-determined to external motivation. Need satisfaction—particularly autonomy and relatedness satisfaction—predicted people’s willingness to get vaccinated. In Israel, autonomy frustration was markedly higher than in the UK, suggesting that people in Israel felt more pressure to get vaccinated. 

There could be several reasons as to why people in Israel are more need-frustrated than people in the UK. Differences in health communication messages, social pressure and other circumstantial, social, and cultural differences between the two countries could all contribute. However, the vaccine passports in Israel, called “green passes”, received considerable backlash and criticism, including several appeals to the Israeli supreme court, with citizens and healthcare experts seeing them as coercion and against individual autonomy and freedom of choice [32]. It seems reasonable to expect, therefore, that vaccine passports would frustrate psychological needs—particularly people’s sense of autonomy—and our data are consistent with this hypothesis. Moreover, we found that the more people felt autonomy frustrated, the more they were against vaccine passports. 

To the extent that vaccine passports do increase psychological need frustration, our data suggest that they might reduce people’s willingness to get vaccinated. A vast body of research showed that the satisfaction of the three psychological needs (autonomy, competence and relatedness) is critical for internalising and maintaining behaviours that improve health and well-being [14,15]. Moreover, frustration of these needs may elicit undesired responses, including disengagement from the activity or doing the opposite of what is requested (oppositional defiance) [19]. Our study extends these findings to vaccination behaviour, showing that people’s willingness to get vaccinated against COVID-19 is related to the satisfaction and frustration of psychological needs around getting vaccinated—particularly their sense of autonomy. For this reason, control measures such as vaccine passports that frustrate psychological needs may have detrimental effects on people’s motivation and willingness to get vaccinated.

Furthermore, if people with frustrated needs do succumb to the pressure to get vaccinated, they are more likely to do so due to external motivation (feeling pressure from others or to satisfy others) rather than due to autonomous identified motivation (wanting to take responsibility over one’s health and understanding the importance of the decision). Although such a possibility would provide some immediate benefits in the form of vaccination rates, it might also produce unintended side effects. For example, as previously mentioned, people might be less willing to receive a “booster” shot or to take a yearly vaccine against new variants. In contrast, if people are autonomously motivated to get vaccinated, sustained adherence to vaccine guidance will be more likely [13,14].

Autonomy-frustrating policies such as vaccine passports may also have long-term public health implications in terms of trust in the health system. People who are amotivated, or who feel pressured are unlikely to build good and trusting relationships with local governments and health authorities—relationships that are crucial for public health adherence and behaviour change to occur [16,33]. Moreover, need frustration can damage people’s well-being, so need-frustrating policies might add to the already heavy burden of the pandemic on people’s mental health [19,34,35]. It is therefore important for governments and policy makers to apply health and risk communication that enhances basic psychological needs, such as creating an autonomy-supportive health care climate and building a caring and trusting relationship with the public (see [16] for full guidelines).

### Strengths and Limitations

It is important to note that, in Israel, domestic vaccine passports are given only to fully vaccinated residents or people who have recovered from COVID-19; this may be more restricted than other passports’ initiatives, such as the “coronapas” in Denmark, where the requirements for a valid coronavirus passport are full vaccination or two weeks since first dose; a negative test taken within the last 72 h; or recent recovery from COVID-19 [9]. This may influence the level of perceived autonomy, and hence the motivation and decision to get vaccinated. In this study, we evaluated attitudes towards domestic vaccine passports for everyday use (e.g., going to restaurants, social events), not for facilitating international travel, which may have different implications and should be examined.

In addition, this study only analysed data from two developed and democratic countries. Although previous research has shown that the satisfaction of basic needs for autonomy, relatedness and competence are essential for optimal functioning across cultures and across individual differences in need strength [28], it is still important to investigate whether our findings are applicable to other countries and cultures.

Furthermore, our study is quantitative in nature, also eliciting qualitative data about attitudes towards vaccine passports, could enhance our understanding of the reasons behind the satisfaction and frustration of needs.

One key strength of this study is that it includes large, representative samples from two different countries (the UK and Israel). Because these countries have advanced vaccination programmes, we were able to investigate the relationships between psychological needs and actual vaccination behaviour in addition to vaccination intentions. A key limitation of this study is the observational design. Although we demonstrated robust relationships between psychological needs and people’s willingness to get vaccinated, we cannot establish causal links between the two. Although it is possible need frustration reduces willingness to get vaccinated, it is also possible that people first decide whether to get vaccinated, and that decision ultimately leads to more or less need frustration. Such a pattern would still be of interest, however. For example, authorities in the US are going to great lengths to encourage people who have chosen not to get vaccinated to change their minds. If people are experiencing need frustration (for example, because of vaccine passports), it is likely to be even more difficult to change their minds [17,34].

## 5. Conclusions

Control measures, such as domestic vaccine passports, may have detrimental effects on people’s autonomy, motivation and willingness to get vaccinated. We should strive to achieve a highly sustainable vaccinated population by supporting individuals’ autonomous motivation to get vaccinated and using messages of autonomy and relatedness. Thus, providing a caring culture and meaningful rationales for a behaviour, rather than applying pressure and external controls.

## Figures and Tables

**Figure 1 vaccines-09-00902-f001:**
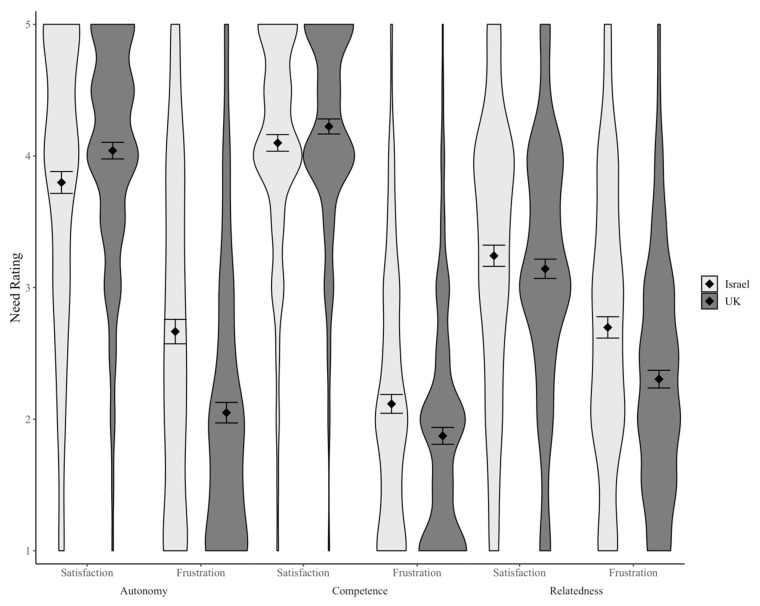
Violin plot displaying the distributions of participants’ need satisfaction and frustration ratings in Israel and the UK. Diamonds represent the cell means and error bars represent the 95% confidence intervals around those means.

**Table 1 vaccines-09-00902-t001:** The items for the 6 basic psychological needs, including their Cronbach’s alpha coefficient of reliability.

Psychological Need Satisfaction and Frustration	Cronbach’s Alpha
Autonomy satisfaction	α = 0.76
I feel [felt] a sense of choice and freedom in the decision to get vaccinated	
I feel [felt] that my decision to get vaccinated reflects what I really want	
Autonomy frustration	α = 0.81
I feel [felt] forced to get vaccinated	
I feel [felt] that I will [would] be ‘punished’ if I didn’t get vaccinated	
Competence satisfaction	α = 0.79
I feel [felt] confident that I could get vaccinated if I wanted to	
I feel [felt] capable of getting vaccinated if I wanted to	
Competence frustration	α = 0.71
I have [had] serious doubts about whether I could get vaccinated if I wanted to	
I feel [felt] that it would be difficult for me to get vaccinated if I wanted to	
Relatedness satisfaction	α = 0.85
I feel [felt] that the official authorities care about me	
I feel [felt] that the official authorities understand [understood] my needs	
Relatedness frustration (α = 0.76)	α = 0.76
I feel [felt] excluded by the official authorities	
I feel [felt] that the official authorities are [were] cold and distant	

**Table 2 vaccines-09-00902-t002:** The items for the 4 types of motivation, including their Cronbach’s alpha coefficient of reliability.

Motivations to Get Vaccinated/Not Get Vaccinated	Cronbach’s Alpha
Identified motivation	α = 0.72
Because I feel that I want to take responsibility for my own health Because I have carefully thought about it and believe this decision is very important for many aspects of my life	
Introjected motivation	α = 0.73
Because I would feel bad about myself if I did [didn’t] get vaccinated	
Because I would feel guilty or ashamed of myself if I did [didn’t] get vaccinated	
External motivation	α = 0.70
Because I feel under pressure from others [not] to get vaccinated	
Because other people would be upset if I do [don’t] get vaccinated	
Amotivation	α = 0.58
I really don’t think about it	
I don’t really care	

**Table 3 vaccines-09-00902-t003:** Demographics and vaccine status of participants from the UK and Israel.

Characteristics	UK	Israel
**n**	681	677
**Age**		
18–29	18%	29%
30–59	53%	56%
60+	29%	15%
**Gender**		
Man	48.5%	48.5%
Woman	51%	51.5%
Non-binary	0.5%	0%
**Education (Highest Level)**		
No formal education	1%	1%
Primary school	0%	1%
Secondary school	34%	41%
Undergraduate degree	43%	38%
Postgraduate degree	22%	19%
**Vaccination Status**		
Unvaccinated	34%	14%
Single dose	41%	4%
Two doses	25%	82%

**Table 4 vaccines-09-00902-t004:** Linear regression of the three psychological needs and willingness to get vaccinated.

Coefficients from Linear Regression			
Term	*β*	t-Statistic	*p*
autonomy_satisfaction	0.17	5.35	<0.001
autonomy_frustration	−0.47	−15.79	<0.001
competence_satisfaction	0.05	1.52	0.128
competence_frustration	0.07	2.66	0.008
relatedness_satisfaction	0.24	7.94	<0.001
relatedness_frustration	0.09	2.88	0.004

**Table 5 vaccines-09-00902-t005:** Logistic regression of the three psychological needs and “decision to get vaccinated”.

Coefficients from Logistic Regression			
Term	Odds Ratio	Statistic	*p*
(Intercept)	−2.83	−3.28	<0.001
autonomy_satisfaction	0.84	5.62	<0.001
autonomy_frustration	1.21	9.08	<0.001
competence_satisfaction	−0.36	−2.19	0.03
competence_frustration	−0.09	−0.68	0.49
relatedness_satisfaction	−1.13	−7.74	<0.001
relatedness_frustration	−0.37	−2.44	0.01

**Table 6 vaccines-09-00902-t006:** Correlations between need frustration and motivation.

Motivation	Autonomy Frustration	Competence Frustration	Relatedness Frustration
External	0.32 **	0.17 **	0.09 *
Amotivation	0.24 **	0.17 **	0.21 **
Identified	−0.46 **	−0.24 **	−0.34 **
Introjected	−0.19 **	−0.05	−0.19 **

* *p* < 0.05, ** *p* < 0.001.

## Data Availability

Data supporting reported results are available upon request to the corresponding author.

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
