# Peer review of "“Vaccine Passports” May Backfire: Findings from a Cross-Sectional Study in the UK and Israel on Willingness to Get Vaccinated against COVID-19"

_vaccines, 2021, doi:10.3390/vaccines9080902_

Round 1
Reviewer 1 Report
In the manuscript by Porat et al, the authors describe the use of psychological need satisfaction and frustration metric system to evaluate the willingness of people from two different countries to get vaccinated against Covid-19. This works provides an interesting analysis on the use of vaccine passports as control measures against Covid-19 and highlights its detrimental effects. The manuscript is well written however some of the tables are hard to read. I will recommend the authors to change the table display format for easy reading.
Table 1: add more columns and delineate rows
Table 2: Add another column to indicate age, gender, education and vaccination status.
Figure 1. indicate the type of graph under the legend: i.e., volcano plot.
Author Response
Response to Reviewer 1 Comments
Point 1: In the manuscript by Porat et al, the authors describe the use of psychological need satisfaction and frustration metric system to evaluate the willingness of people from two different countries to get vaccinated against Covid-19. This works provides an interesting analysis on the use of vaccine passports as control measures against Covid-19 and highlights its detrimental effects. The manuscript is well written however some of the tables are hard to read. I will recommend the authors to change the table display format for easy reading.
Table 1: add more columns and delineate rows
Table 2: Add another column to indicate age, gender, education and vaccination status.
Figure 1. indicate the type of graph under the legend: i.e., volcano plot.
Response 1: We thank the reviewer for the helpful feedback.
• Table 1 - we have now added columns to Table 1 and divided the table to two
separate tables (Table 1 for the 6 basic psychological needs items and Table 2 for the 4 motivation items). We believe this is much clearer now.
• Table 2 (changed now to Table 3) already includes rows indicating age, gender,
education and vaccination status (we have now highlighted the titles to improve
clarity). We tried changing the rows to columns but this made the table too wide. If, however, the current formatting is confusing or if the reviewer wished us to add additional information, we are happy to do so.
• Figure 1 - we have expanded the figure caption to read: “Violin plot displaying the distributions of participants’ need satisfaction and frustration ratings in Israel and the UK. Diamonds represent the cell means, error bars represent the 95% confidence intervals around those means”.

Reviewer 2 Report
First of all, I am grateful for the opportunity to review this paper. COVID-19 is an ongoing pandemic that has resulted in global health, economic and social crises. Actually, the vaccination campaign is the first method to counteract the COVID-19 pandemic; however, sufficient vaccination coverage is conditioned by the people’s acceptance of these vaccines especially by health professionals. In this context, the paper under review is aimed at understanding people’s willingness and motivation to get vaccinated and how vaccine passports might affect these needs.
The article is interesting and may provide important information for public health, but it must be improved.
Introduction: The authors should make it clear about what is the gap in the literature that is filled with this study? First of all, the general acceptance of the COVID-19 vaccine (before the introduction of the vaccine passport) must be better discussed (refer to Gallè, F. et al Knowledge and Acceptance of COVID-19 Vaccination among Undergraduate Students from Central and Southern Italy. Vaccines 2021, 9, 638). What is the contribution of the study to the literature? What are the implications of the study?
Methods: A pilot study was conduced, but how was the questionnaire validated (face validity, intelligibility, reliability)? Moreover, it is not clear how the Authors choose the two counties and what the reason of this comparison is. Statistical analysis: I suggest to insert a measure of the magnitude of the effect for the comparisons.
Ethical Issue: although an anonymous questionnaire is used, an ethical approval is necessary. An ethical committee should approve the study protocol, and a reference number should be reported.
Discussion: I also suggest expanding. Emphasize the contribution of the study to the literature, the implications and recommendations based on previous experience also in other population groups also discussing effectiveness of the information strategy (refer to Gallè, F. et al Knowledge and Acceptance of COVID-19 Vaccination among Undergraduate Students from Central and Southern Italy. Vaccines 2021, 9, 638). Limits section must be better described.
English must be improved.
Author Response
Response to Reviewer 2 Comments
Point 1: Introduction: The authors should make it clear about what is the gap in the literature that is filled with this study? First of all, the general acceptance of the COVID-19 vaccine (before the introduction of the vaccine passport) must be better discussed (refer to Gallè, F. et al Knowledge and Acceptance of COVID-19 Vaccination among Undergraduate Students from Central and Southern Italy. Vaccines 2021, 9, 638). What is the contribution of the study to the literature? What are the implications of the study?
Response 1: Thank you for this feedback. We have now edited and elaborated the Introduction to:
• Highlight the gap in the literature that this study addresses (see rows 65-87).
• Add information about the general acceptance of Covid-19 vaccine at the start of the vaccination programme (referring to Gallè et al. 2021) (see rows 34-63).
• Highlight the contribution and implications of the study (see rows 103-144 and 154-162. This was also highlighted in the ‘Discussion’).
Point 2: Methods: A pilot study was conducted, but how was the questionnaire validated (face validity, intelligibility, reliability)? Moreover, it is not clear how the Authors choose the two counties and what the reason of this comparison is. Statistical analysis: I suggest to insert a measure of the magnitude of the effect for the comparisons.
Response 2: Thank you for this comment.
• The scales were adapted from validated measures – see section 2.3 ‘Main outcome measures’. For example, to measure psychological need satisfaction and frustration, we adapted 12 items from the Basic Psychological Need Satisfaction and Frustration Scale (BPNSNF) (Chen et al., 2015) to the context of getting vaccinated, with two items assessing each of the six constructs – autonomy satisfaction, autonomy frustration, relatedness satisfaction, relatedness frustration, competence satisfaction, and competence frustration. We also used the pilot study to check that participants did not find any of the questions confusing. The Hebrew translation was based on Benita et al. (2020). We have analysed and reported the internal consistency (Cronbach’s alpha) of the subscales for the 6 basic psychological needs and the 4 types of
motivations (see Tables 1 and 2).
• In the ‘Design and Setting’ section (under ‘Materials and Methods’) we have added the reason for selecting Israel and the UK for our study.
Statistical analysis – we are not entirely clear on what the reviewer is looking for, here.
We report the magnitude of the differences between the UK and Israel in the form of mean differences, which convey the raw magnitude of the effects in a way that is interpretable in the context of the scales themselves (for example, a mean difference of 1 on autonomy frustration would represent a difference of one Likert point on the scale between the two countries). But if the reviewer thinks additional information about the effect sizes (e.g. Cohen’s d) is important, we are happy to add it.
Point 3: Ethical Issue: although an anonymous questionnaire is used, an ethical approval is necessary. An ethical committee should approve the study protocol, and a reference number should be reported.
Response 3: Ethical approval for the study was given by both Imperial College London and the Technion. This is written under “Institutional Review Board Statement” (The Technion approval statement was now added).
Point 4: Discussion: I also suggest expanding. Emphasize the contribution of the study to the literature, the implications and recommendations based on previous experience also in other population groups also discussing effectiveness of the information strategy (refer to Gallè, F. et al Knowledge and Acceptance of COVID-19 Vaccination among Undergraduate Students from Central and Southern Italy. Vaccines 2021, 9, 638). Limits section must be better described.
Response 4: Thank you for this comment. We have now edited and elaborated the Discussion to:
• Emphasise the contribution of the study to the literature (see rows 380-391), the implications and recommendations (see rows 393-412).
• Expand the limitations section (see additional limitations in rows 528-535).
Point 5: English must be improved.
Response 5: We have edited the manuscript to improve the clarity and are confident the English in the revised version is improved.

Round 2
Reviewer 2 Report
The manuscript was significantly improved and it is now suitable for publication. Please a final check to the reference style is necessary (eg. reference n. 4 is missing "et al." among the authors.